# Advances in the Treatment of Pediatric Brain Tumors

**DOI:** 10.3390/children10010062

**Published:** 2022-12-27

**Authors:** Edwin S. Kulubya, Matthew J. Kercher, H. Westley Phillips, Reuben Antony, Michael S. B. Edwards

**Affiliations:** 1Department of Neurosurgery, University of California Davis, Sacramento, CA 95817, USA; 2Department of Neurosurgery, Children’s Hospital of Pittsburgh, University of Pittsburgh Medical Center, Pittsburgh, PA 15224, USA; 3Department of Pediatrics, University of California Davis, Sacramento, CA 95817, USA

**Keywords:** pediatrics, brain tumors, glioma, medulloblastoma, ependymoma, immunotherapy, targeted therapy, laser interstitial therapy

## Abstract

Pediatric brain tumors are the most common solid malignancies in children. Advances in the treatment of pediatric brain tumors have come in the form of imaging, biopsy, surgical techniques, and molecular profiling. This has led the way for targeted therapies and immunotherapy to be assessed in clinical trials for the most common types of pediatric brain tumors. Here we review the latest efforts and challenges in targeted molecular therapy, immunotherapy, and newer modalities such as laser interstitial thermal therapy.

## 1. Introduction

Pediatric brain tumors are the leading cause of cancer-related death in childhood [1,2]. They constitute a heterogenous group with varying developmental origins, genomic profiles, therapeutic options, and outcomes. Tumors seen in younger children (less than three years) often have an embryonal origin, while brain tumors in older children are more likely to derive from glial cells [3,4]. Advances in surgery, neuro-oncology, neuroradiology and radiation oncology have improved patient survival for some tumor types such as low-grade gliomas and medulloblastomas. However, the prognosis for patients with diffuse midline gliomas, other high-grade gliomas (HGGs), and most recurrent pediatric brain tumors remains poor since current therapeutic strategies are unable to extend survival by more than a few months in most patients [5]. Most high-grade pediatric brain tumors require treatment with intensive chemotherapy regimens and focal or craniospinal irradiation in addition to surgery, which can have devastating long-term effects on development and cognition. Improving survival rates in this patient cohort has been the primary focus of most pediatric-cancer treatment consortia such as the Children’s Oncology Group, but an equally important aim has been to minimize the immediate and long-term side effects of chemotherapy and ionizing radiation (Table 1). Since the publication of the seminal study by Pomeroy et al. in 2002, in which central nervous system (CNS) embryonal tumor outcomes were successfully predicted using gene expression patterns, a concerted effort has been made to elucidate the molecular mechanisms driving the origin and progression of pediatric brain tumors [6]. This has resulted in improvements in diagnostic accuracy, identification of potential therapeutic targets, and in some cases, implementation of targeted therapies which achieve tumor-volume control with minimal off-target effects. In this review, we aim to review the latest approaches towards pediatric brain tumors centered around targeted molecular therapy, immunotherapy, and newer modalities such as laser interstitial thermal therapy.

## 2. Innovations in Diagnosis

The clinical presentation of pediatric brain tumors is predicated on the anatomic location of the tumor, the presence of regional mass effect, associated hydrocephalus, and the patient’s age. Infants may present with nonspecific symptoms such as irritability, macrocephaly, growth failure, and weight loss, along with tumor-related symptoms such as headache, nausea, and vomiting [7]. Patients with supratentorial tumors usually present with seizures or signs of elevated intracranial pressure. Posterior fossa tumors, which comprise almost half of all pediatric brain tumors, usually have headache, nausea, vomiting, headache, abnormal gait, and abnormal coordination as presenting signs. These tumors often compress the fourth ventricle or block cerebrospinal fluid (CSF) pathways, causing an obstructive hydrocephalus and subsequent symptoms of elevated intracranial pressure. Brainstem tumors present with abnormal gait and coordination as well as cranial nerve palsies, while suprasellar tumors present with symptoms and signs of (endocrine) pituitary dysfunction.

### 2.1. Imaging

From diagnosis and surgical planning to surveillance and assessment of treatment response, imaging is essential to the management of pediatric brain tumors. Magnetic Resonance Imaging (MRI) is the gold standard when it comes to assessing brain tumors. Standard sequences include pre- and post-contrast T1-weighted, T2-weighted, fluid-attenuated inversion recovery (FLAIR), and diffusion-weighted sequences [8]. Diffusion tensor imaging (DTI) can help assess the location and integrity of white matter tracts preoperatively [9]. Postoperatively, magnetic resonance spectroscopy (MRS) and perfusion-weighted imaging (PWI) can help differentiate tumor recurrence from treatment-induced change [10]. The Response Assessment in Pediatric Neuro-Oncology (RAPNO) working group was established to address challenges in assessing response in children with pediatric brain tumors and have published imaging guidelines for specific tumor types [11,12,13].

In the last decade, technological advances in diagnostic imaging have enabled us to regard MRI images as reflections of tumor biology, metabolism, gene expression, angiogenesis, and proliferation. The emerging field of radiogenomics aims to define the relationships between noninvasive imaging features (radiophenotypes) and molecular/genomic features (molecular phenotypes). Genomic alterations in many brain tumors result in reprogramming of tumor-cell metabolism, including glucose, amino acid, and lipid metabolism, which can be visualized using noninvasive clinic imaging [14]. In addition, machine learning is also being used to enhance the accuracy of radiological diagnosis in posterior fossa tumors and even medulloblastoma subgroups [15,16]. The more data is collected, the better noninvasive imaging diagnosis can become.

### 2.2. Biopsy

Biopsy and tumor-tissue analysis is required for the final pathological diagnosis of brain tumors. Midline brain tumors or brainstem tumors are now biopsied more often for definitive molecular diagnosis and clinical trials. Some germ-cell tumors secrete tumor markers (Alpha-fetoprotein (AFP) and beta-human chorionic gonadotropin (bhCG)) and can be diagnosed by identification of those markers in CSF and blood. At present, several research groups are looking at proteomic analysis of CSF to develop and identify protein biomarkers associated with specific pediatric brain tumors [17,18]. Novel techniques, such as isolating cell-free DNA (cf-DNA) in CSF, blood, and urine, are being studied so that minimally invasive liquid biopsies will enable easy diagnosis of CNS tumors and serve as a surveillance tool for relapse.

### 2.3. Surgery

The aims of surgery are to provide tissue diagnosis, perform maximal safe tumor resection, alleviate mass effect, and restore CSF-flow abnormalities. Workhorse craniotomies, including traditional pterional and suboccipital craniotomies, are commonly employed. Contemporary approaches and techniques have been employed to improve patient outcomes and decrease morbidity. Stereotaxis has streamlined minimally invasive approaches to biopsy intracranial lesions, especially those in subcortical locations. Transcranial magnetic stimulation, coupled with functional MRI (fMRI), allows for preoperative motor and language mapping for children in whom direct cortical stimulation during awake surgery is not feasible [19,20].

Furthermore, new technology has developed that equips surgeons with novel ways to approach and resect a plethora of brain tumors. For instance, minimally invasive flexible and rigid endoscopy has made it easier to biopsy ventricular and pineal-region tumors. Transnasal endoscopy has been shown to be safe and effective in children and provides advantages for skull-base or sellar lesions, such as pediatric craniopharyngiomas and germ-cell tumors [21]. Endoscopy can also be used to aid in traditional microscopic surgeries and keyhole approaches. Endoscopes can provide better visualization of the structures that could be hidden when a microscope is used [22]. Ultrasonic aspirators, including neuroendoscopic ultrasonic aspirators, have enhanced tumor debulking techniques, especially for firm lesions [23]. All these tools help achieve much safer and maximal resections.

### 2.4. Molecular Profiling

Research into the molecular mechanisms underlying pediatric brain tumor causation and maintenance has resulted in the classification of various tumor types into mechanistic subgroups based on genomic profiling, with robust correlations between specific or combinations of genomic alterations and clinical outcomes. This insight into molecular mechanisms has resulted in increased accuracy and reliability of tumor diagnosis, has revealed potential therapeutic targets, and provides a rational basis to either augment or de-intensify therapy. Molecular sub-classification has been integrated into the fifth edition of the World Health Organization (WHO) Classification of Tumors of the Central Nervous System, which was published in 2021, with grading of certain tumors based on genomic alterations [24]. Interrogating pediatric brain tumors with next-generation sequencing allows for targeted high-throughput sequencing of tumor tissue for genomic alterations, mutations, copy number variations, translocation, and fusions of multiple genes and has become the standard of care for pediatric brain tumors in most treatment centers [25].

## 3. Treatment Advances

### 3.1. Targeted Therapy

Targeted therapy, sometimes called personalized or precision medicine, is an emerging field of therapeutics in which druggable receptors or canonical driver molecular pathways are identified in tumor cells, and drugs are either developed or repurposed to inhibit the function of the target, thereby limiting tumor growth. This therapeutic strategy is applicable most effectively to tumors which have single well-defined targets with a paucity of redundant escape pathways, such as BRAF V600-altered low-grade gliomas. At the current time, the Children’s Oncology Group has several active trials testing the efficacy and tolerability of different targeted agents. Challenges facing targeted therapy include the ability of the drug to cross the blood–brain barrier, distinct tumor-cell populations, or multiple driver alterations within the same tumor resulting in tumor-cell escape mechanisms [26,27]. Intratumoral heterogeneity remains a key factor in therapeutic resistance and treatment failure and is associated with a poor prognosis [28].

#### 3.1.1. Low-Grade Gliomas

Low-grade gliomas (WHO grade I and II) are a heterogenous group that make up the largest subset of pediatric brain tumors. The primary cell types of these tumors are varied and include neuronal, mixed glioneuronal, astrocytic, oligodendroglial, or oligoastrocytic cells. Often, pediatric low-grade gliomas are caused by alterations in the signaling pathway of mitogen-activated protein kinase (MAPK), including the BRAF mutation or fusion, the FGFR1 mutation or structural rearrangement, the NF1 Mutation, and the RAF and NTRK family fusions [29]. In diffuse low-grade gliomas, the MAPK pathway is frequently altered [24]. BRAF-KIAA fusions are seen in cerebellar and optic pathway pilocytic astrocytomas, while BRAF mutations are common in ganglioglioma, pleomorphic astrocytoma, and cerebral pilocytic astrocytoma [2]. In some patients, gross total resection is feasible and curative. In others, recurrence is frequently seen and chronic management is necessary with a focus on minimizing treatment toxicity [30].

One of the biggest successes thus far in molecular therapy has been in low-grade gliomas with targeted therapy helping reduce cytotoxic chemotherapy regiments. Evorolimus, an mTOR-pathway inhibitor, is highly effective in the treatment of subependymal giant-cell astrocytoma in tuberous sclerosis [31]. The MEK1/2 inhibitor, Selumetinib, has shown promise in the treatment of new and recently diagnosed BRAF KIAA-duplicated low-grade gliomas [32,33]. Due to these results, studies are now underway to compare the efficacy of Selumetinib with the efficacy of the long-standing standard-of-care therapy regimen for pediatric low-grade gliomas, Carboplatin/Vincristine (in patients with NF1), and to compare Vinblastine monotherapy to a combination of Vinblastine and Selumetinib in recurrent or progressive pediatric low-grade gliomas (in Non-NF1, Non-TSC patients) [34]. Another MEK inhibitor, Trametinib has also been effective in patients with recurrent or progressive low-grade gliomas [35,36]. The BRAF-pathway inhibitors Dabrafenib and Vemurafenib have been used as upfront monotherapy in patients with BRAFV600-mutated low-grade gliomas as well as in combination with MEK inhibitors. Currently the Pacific Pediatric Neuro-Oncology Consortium (PNOC) is testing the oral Pan RAF inhibitor DAY101 against recurrent or progressive BRAF-altered pediatric low-grade gliomas (NCT04775485). The main limitation in the use of targeted therapy in low-grade gliomas that has been observed is that tumor progression is seen in a significant number of patients when therapy is held.

#### 3.1.2. High-Grade Gliomas

High-grade gliomas (WHO Grade IV) have the highest mortality rate of all pediatric brain tumors, with five-year survival rates less than 20% despite maximal therapeutic interventions [37]. Pediatric high-grade gliomas constitute a genomic group separate from adult high-grade gliomas given the high frequency of driver mutations in the histone 3 variant H3.3 [5,38,39]. The WHO Classification of Tumors of the Central Nervous System, Fifth Edition, now describes diffuse midline glioma, H3K27-altered; diffuse hemispheric glioma, H3 G34-mutant; diffuse pediatric-type high-grade glioma, H3-wildtype and IDH-wildtype; and Infant-type hemispheric glioma [24]. Diffuse midline glioma, H3K27-altered includes the previously known diffuse intrinsic pontine glioma and midline unilateral thalamic glioma. Diffuse hemispheric glioma, H3G4-mutant arise in older children and young adults [40]. Diffuse pediatric high-grade glioma, H3 wild type and IDH wild type are further divided into those with enriched MYCN, PDGFRA, or EFGR amplification [41]. Infant-type hemispheric glioma contain receptor tyrosine kinase gene fusions, including ALK, NTRK1/2/3, ROS1, and MET4 [40].

The insight we have gained into the genomic drivers of pediatric high-grade gliomas has resulted in several clinical trials testing targeted agents against this group of tumors. Histone deacetylase (HDAC) inhibitors, such as panobinostat, have been used to target the H3K27 mutant but are still in preclinical testing [42]. Therapy has been limited due to limited penetration of the blood–brain barrier, resistance to HDAC inhibition and other adverse effects [34]. At present, Fimepinostat, or CUDC-907, is a dual PI3k and histone deacetylase (HDAC) inhibitor and is also being studied in high-grade glioma and medulloblastoma (NCT02909777) (NCT03893487). Larorectinib has been used with some success to treat NTRK-fusion positive pediatric HGGs [37]. The PARP inhibitor BGB-290 in combination with Temozolomide is being tested in the treatment of IDH 1/2 mutated high-grade gliomas (NCT03914742). The dopamine receptor D2 (DRD2) and caseinolytic protease proteolytic subunit (CLpP) antagonist Onc201 is being tested in combination with Panabinostat and the PI3K/AKT/mTOR inhibitor Paxalisib in children and adults with several subgroups of brain and spinal cord diffuse midline gliomas at diagnosis, post radiation, and during progression (NCT05009992).

Monoclonal antibodies, which target surface proteins, have been used to treat solid malignancies but have also been limited in brain tumors. Bevacizumab, a humanized monoclonal antibody which binds circulating vascular endothelial growth factor (VEGF) and prevents this angiogenic cytokine from binding to its receptor, is used in adult progressive glioblastoma multiforme [43]. Adding bevacizumab to the treatment of pediatric high-grade gliomas did not improve survival, limiting its application to the treatment to radiation necrosis and peritumoral edema [44]. However, in recent years, Bevacizumab has been used increasingly in the management of low-grade gliomas, in which it provides good tumor control and improvement of vision in patients who present with visual impairment secondary to low-grade gliomas of the visual tract [44,45].

The PI3K/AKT/mTOR pathway promotes cell survival and growth and is activated in several cancers [46]. Alterations in this pathway have been seen in pediatric brain tumors and several clinical trials are underway to test agents which target downstream mTOR signaling [26]. The PI3K/mTOR inhibitor Samotolisib is currently being tested in pediatric patients with a variety of recurrent brain tumors, including high-grade glioma, ependymoma, and medulloblastoma (NCT03213678).

#### 3.1.3. Medulloblastoma

Medulloblastoma is the most frequent malignant (WHO grade IV) brain tumor in children. Molecular subgroups of medulloblastoma have been well defined and now include wingless (WNT)-activated, sonic hedgehog (SHH)-activated and the NON-WNT, NON-SHH medulloblastoma, which includes group 3 and group 4 medulloblastoma [40,47]. These have been further delineated into genetic subtypes [34]. Regardless of subtype, the treatment of medulloblastoma consists of maximal safe resection that leaves no more than 1cmx1cm residual mass. This is followed by craniospinal radiation with a boost to the tumor bed, followed by several cycles of chemotherapy.

WNT-activated medulloblastomas account for nearly 10% of medulloblastomas and have the best prognosis, with greater than 95% five-year survival [48,49]. They are often located in the cerebellar midline and can involve the cerebellar peduncles and brain stem. WNT tumors are identified by positive nuclear beta-catenin staining due to CTNNB1 mutations and monosomy 6 [49]. Due to the high survival rates seen in patients with WNT tumors, clinical trials have focused on risk adaptive therapies to reduce chemotherapy and craniospinal radiation doses [50]. Studies are exploring reduced doses of craniospinal RT at 18Gy and chemotherapy (NCT02724579), but some early cases demonstrated an increased risk of relapse. One chemotherapy-only study was terminated due to a high rate of relapse in study participants (NCT02212574). Therefore, one current study is testing how a reduction of CSI to 18gy plus a tumor-bed boost of 36Gy followed by adjuvant systemic therapy will affect patient outcomes (NCT04474964). WNT signaling has been shown to inhibit the SHH pathway in SHH medulloblastomas, and WNT agonists are also being explored as potential tumor suppressors in group 3 and 4 medulloblastomas [51,52].

SHH medulloblastoma is characterized by mutations along the SHH signaling pathway, including SUFU, PCTH, SMO, and GLI [53]. They account for about 30% of medulloblastomas [40]. SHH medulloblastomas have been further defined by the presence or absence of a mutation in the TP53 tumor suppressor gene. SHH-activated tumors with mutant TP53 have a poorer prognosis than SHH-activated TP53 wild type tumors [40]. TERT promoter mutations are also present in 40% of SHH medulloblastomas and in most adult cases [49]. There are also four more subtypes of SHH medulloblastoma: alpha, beta, gamma, and delta. Targeted therapy for some SHH-activated medulloblastoma involves the sonic hedgehog inhibitors vismodegib and sonidegib, which bind smoothened (SMO) to prevent nuclear transcription of SHH target genes (because Vismodegib/Sonidegib strategy is applicable only for tumors which do not have an alteration downstream of SMO). SHH Inhibition causes growth and development impairment due to premature bone-growth-plate fusion, but partial recovery of growth was demonstrated upon long-term follow-up [50,54,55]. CK2, a downstream kinase and driver of hedgehog, is another potential molecular target [56]. In addition, the CK2 inhibitor CX-4945 was shown to sensitize medulloblastoma cells to temozolomide treatment [57].

Non-WNT/non-SHH medulloblastomas are categorized into Group 3 and Group 4, which then have eight further subtypes. Group 3 medulloblastomas tend to occur in infants and children and have the worst prognosis with five-year survivals rates of 50% [58]. Group 3 medulloblastomas are subdivided into low-risk and high-risk groups, with the 10-year overall survival in the Group 3 low-risk group being about three times that of the high-risk group. The Group 3 low-risk group of medulloblastomas is characterized by MYC amplification [48]. Treatment in young children is focused on adjuvant high-dose chemotherapy with autologous stem-cell rescue to delay radiation therapy. Group 4 medulloblastomas are the most common and occur in older children and adolescents. Genetic alterations have stratified prognosis by defining a high-risk group characterized by the presence of isochrome 17q and a low-risk group defined by the loss of chromosome 11 and MYCN amplification. The low-risk group has a 10-year survival rate of 72%, which is twice that of the high-risk group [59,60]. Further studies are needed to test risk-adaptive therapies for these groups.

#### 3.1.4. Pineoblastoma

Pineoblastoma is a rare, aggressive embryonal tumor of the pineal gland with poor long-term survival despite multi-modal surgical, chemotherapeutic, and radiotherapy treatments. Subgroups of pineoblastoma are starting to be defined as groups 1, 2, 3, RB, and MYC [61,62]. Group 1 and 2 tumors express homozygous loss-of-function alterations in genes DICER1, DROSHA, and DGCR8. PB-Rb tumors show RB1 loss with gain of *miR-17/92* and include cases of trilateral retinoblastoma. PB-MYC is characterized by MYC activation. Groups 1–3 tend to occur in older children and have better survival, while PB-RB and PB-MYC occur in younger children, with poor survival [61]. No clinical trials exist for the sole treatment of pineoblastoma. Historically, treatment has been grouped with trials for medulloblastoma, given their histological similarity [63]. There is some preclinical data supporting testing the efficacy of the tricyclic antidepressant Nortriptyline in combination with Gemcitabine to treat Rb-mutated pineoblastoma by inhibition of lysosomal function [64].

#### 3.1.5. AT/RT

Atypical rhabdoid teratomas (AT/RT) are rare but lethal embryonal tumors that occur in young infants, with median survival between 6 and 11 months [65]. Histologically, these tumors have been misdiagnosed as medulloblastoma or PNET [66]. AT/RTs are now defined by mutations in INI1 or BRG1 [67]. AT/RTs are characterized by biallelic loss-of-function alterations in the SMARCB1/INI1 tumor-suppressor gene on chromosome 22q [40]. Rare cases with intact SMARCB1 harbor mutations in SMARCA4/BRG1 [68]. Inactivation of these proteins leads to tumorigenesis. Using DNA methylation and gene-expression profiling, three molecular subgroups have been identified: TYR, SHH, and MYC [68,69]. AT/RT-TYR (Group 2) occurs predominantly in the posterior fossa; it has point mutations and focal SMARCB1 deletions and overexpresses tyrosinase. ATRT-SHH (Group1) can occur in both supratentorial and infratentorial locations; it has focal SMARCB1 deletions and overexpresses SHH. ATRT-MYC occurs most commonly in the supratententorium, has broad SMARCB1 deletions, and overexpresses MYC and HOX cluster genes [69].

Current treatment involves surgery, systemic and intrathecal chemotherapy, and craniospinal irradiation. High-dose chemotherapy with autologous bone marrow rescue with or without radiation has also been used, but overall two-year survival is only 15–45% [40,70]. Targeted therapies are focused on inhibitors of Aurora Kinase A (Alisertib, NCT02114229), CDK4/6 (Ribociclib NCT03434262), and EZH2 (Tazemetostat NCT02601937) [70]. Preclinical investigation of the proteosome inhibitor marizomib (MRZ) is also underway. MRZ can cross the blood–brain barrier and induce cancer-cell death by increasing intracellular reactive oxygen species [70].

#### 3.1.6. Ependymoma

Ependymomas constitute about 5% of childhood brain tumors and occur more frequently in the posterior fossa [71]. Ependymomas are classified by location and genetic and epigenetic alterations. Supratentorial ependymomas are comprised of C11orfn95/ZFTA (former *RELA*) fusion positive and YAP1 fusion-positive ependymoma [40,72]. ST-YAP1 tumors occur mostly in children compared to ST-ZFTA tumors [73]. Inhibitors of RELA pathways or NF-κB signaling pathways are being explored for potential therapy [74].

Posterior fossa ependymomas are comprised of group A (PFA) and group B (PFB) [75]. PFA ependymomas do not have recurrent mutations but demonstrate H3K27 methylation, with higher levels seen in PFB ependymomas [72]. PFA ependymomas are much more prevalent and often occur in the lateral cerebellum in infants and have a distinct DNA-methylation profile with hypermethylated loci converging on targets of polycomb repressor complex 2 (PCR2) [48]. PFB ependymomas tend to occur in the midline cerebellum and more frequently occur in older children and adults [76]. Treatment involves maximal safe resection followed by focal conformal radiation to the postoperative tumor field, except for infants less than one year of age who do not receive radiation [40,71]. Subtotal resection of PFA ependymomas has a very poor prognosis even with adjuvant radiotherapy [71]. PFB ependymomas have a significantly better prognosis and, in some situations, may require surgery alone [76]. The efficacy of post-radiation chemotherapy in treating ependymoma in various settings is still being investigated, and chemotherapy is used for high-risk, subtotally resected, or recurrent disease. There is no current targeted therapy for posterior fossa ependymomas, but targeting H3K27 may be of value. Inhibition of histone lysine methylation decreased survival of PFA ependymoma cell lines [77]. Currently, there is no effective treatment for recurrent ependymoma, and despite re-irradiation and repeat surgical resection, most patients experience disease progression. The PNOC consortium is at present conducting a phase 1 clinical trial testing the CD47 antagonist Magrolumab against recurrent or progressive ependymoma (NCT05169944).

### 3.2. Immunotherapy

Driven by success in some adult cancers, there has been an increase in clinical trials utilizing various forms of immunotherapy for both adult and pediatric CNS tumors. These tumors are also described as immunologically “cold” [1]. A limiting factor is each tumor’s unique microenvironment (TME). In adult high-grade glioma, the TME contains immunosuppressive cytokines (TGF-B, IL-10), chemokines, tumor-associated macrophages/microglia (GAMs), and myeloid-derived suppressor cells to limit therapy [37]. The TME in pediatric high-grade glioma is thought to contain even fewer immune cells and inflammatory factors, as well as rare antigen-presenting cells, creating a challenge for immunotherapy. Risks of immunotherapy or immune-related adverse events include myasthenia gravis, encephalitis, peripheral neuropathy, and meningitis. Combining these immunotherapy modalities may help overcome some of these barriers and improve outcomes.

#### 3.2.1. Immune Checkpoint Inhibitors

Immune checkpoint inhibitors have been studied against adult brain tumors, and several clinical trials are underway to utilize anti-PD1 and CTLA-4 inhibitors in pediatric high-grade glioma [1]. Some success has been seen in hypermutated high-grade gliomas on a background of mismatch repair deficiency [78]. Many of these agents are used in combination with radiotherapy, making it difficult to determine the relative beneficial effect of each modality. Immune checkpoint inhibitors also rely on an immunocompetent host and may be limited by ongoing chemotherapy and steroid use [1]. ICIs have not shown any survival benefit in pediatric high-grade glioma [73].

#### 3.2.2. Adoptive Cellular Therapy

Chimeric antigen receptor (CAR) T-cell therapy utilizes autologous T-lymphocytes that are genetically modified to express CAR and unlock T-cell cytotoxicity against specific tumor-cell antigens (TAA) [1,37]. Given this therapy’s success treating pediatric leukemia, there is optimism that target antigens found on pediatric brain tumors can also be treated using this therapy [79]. GD2 and B7-H3 are tumor antigens found highly expressed in DIPG [37]. GD2-targeted CAR T cells are being studied in HRK27M diffuse midline glioma (NCT04196413). Early results showed clinical and radiographic improvement in three out of four patients, with manageable toxicities [80]. Another phase 1 study evaluating HER2 (CAR) T cells in recurrent and progressive ependymoma is underway (NCT04903080). B7-H3 CAR-T cell trials have also begun in pediatric HGG (NCT04897321). Preclinical studies have also targeted GPC-2, which is found in medulloblastoma and high-grade glioma [81].

A limitation of CAR-T therapy is the idea of “antigen escape” in that the target antigen may be downregulated or the selection of target-negative subclones [37]. Pediatric HGGs are heterogeneous and contain subpopulations of tumor cells with different antigens. For this reason, in adult high-grade glioma, CAR-T cells were designed to target multiple antigens such as HER-2, IL-13Ralpa2, and EphA2 simultaneously [82,83]. To combat the TME, T-cells redirected for universal cytokine-mediated killing (TRUCKs) cells were developed with immune stimulatory cytokines that improve CAR-T cell expansion and persistence [84]. Preclinical studies have also demonstrated efficacy of CAR-T cell therapy delivered directly into the cerebrospinal fluid for metastatic medulloblastoma and PFA ependymoma [85].

#### 3.2.3. Antibody-Mediated Immunotherapy

Monoclonal antibodies targeting the tyrosine kinase receptor HER-2 were extremely successful in the treatment of breast cancer. HER-2 is highly expressed in CNS cancer stem cells and in medulloblastoma [37]. However, this therapy is limited by the antibodies’ ability to cross the blood–brain barrier. Bi-specific T-cell engagers (BiTEs) were also developed to target different antigens and are a class of monoclonal antibodies. BiTEs can help initiate a larger immune response and are being used in combination with adoptive cellular therapy [1]. Monoclonal antibodies can also deliver radio-immunotherapy or immunotoxins in conjunction with convection-enhanced delivery [86].

#### 3.2.4. Cancer Vaccines

Cancer vaccines come in the form of dendritic cells, peptide, nucleic acid, or viral vectors and work to mount an immune response against tumor-specific epitopes [1]. An H3.3K27M-specific vaccine was shown to be safe in a Phase 1 Trial of 19 patients with diffuse midline glioma, with further efficacy results still pending (NCT02960230) [87]. An early phase clinical trial being conducted by the Pediatric Brain Tumor Consortium is testing SurVaxM, a survivin long-peptide vaccine previously trialed in glioblastoma, in children with medulloblastoma, high-grade glioma, ependymoma, and DIPG (NCT04978727). Children may also be better at mounting a T-cell immune response in response to vaccine therapy versus adults with thymic atrophy [88].

#### 3.2.5. Oncolytic Viral Therapy

Oncolytic viral therapy involves genetically modified viral agents designed to selectively replicate in tumor cells [89]. Poliovirus (NCT03043391), measles virus (NCT02962167), wild-type reovirus combined with sargramostim (NCT02444546), cytomegalovirus RNA-pulsed dendritic cells (NCT03615404), and HSV G207 (NCT03911388 and NCT0245784) are being tested in recurrent or refractory medulloblastoma [90]. HSV-G207 has been used in pediatric high-grade glioma with radiation (NCT02457845) [91] and DNX-2401 Virus in DIPG (NCT03178032) [92]. Pelareorep, a virus that can replicate in *ras* mutant cells, was given with GM-CSF (sargramostim) to six pediatric patients with recurrent high-grade tumors and was well tolerated. Further studies are needed to determine efficacy of viral therapies.

### 3.3. Convection Enhanced Delivery (CED)

CED is a method of helping drugs or antibodies bypass the blood–brain barrier and get delivered directly to the tumor in high concentrations. In CED, a cannula is placed within the interstitial spaces of the brain, with its tip within the tumor target. Bulk flow driven by fluid gradients is induced by controlled infusions of drug or therapy and is able to spread to target tissues [93]. CED has been studied for DIPG with long-term data emerging from a dose-escalation study of a radioimmunoconjugate showing safety and an increase in survival of 3–4 months compared to historical controls [86,94].

### 3.4. Laser Interstitial Thermal Therapy (LITT)

Laser Interstitial Thermal Therapy (LITT) is a minimally invasive surgical procedure in which a LASER fiber catheter is stereotactically positioned with its tip inside the tumor mass. Nonionizing laser light is introduced through this catheter to increase the temperature of the targeted tumor volume during concomitant magnetic resonance imaging to guide the extent of ablation. The photons heat the target tissue either directly through absorption or indirectly through scatter into the adjacent tissue, which creates DNA and protein denaturation, leading to cell death. At temperatures above 60 degrees Celsius, cell death is irreversible; between 46 to 60 degrees Celsius, time to cell death is inversely proportional to temperature and the time spent at that temperature [95]. The two FDA approved systems are the Visualase system (Medtronic) and the NeuroBlate system (Monteris Medical) [96]. A benefit to this approach is the ability to take a surgical biopsy at the time of catheter placement for diagnostic purposes. LITT is increasingly being used in adult treatment of high-grade glioma, brain metastases, and deep-seated lesions. In pediatrics, its role is expanding from epilepsy treatment to tumors. In the pediatric population, this minimally invasive treatment is associated with limited hair clipping, small surgical incisions, and shorter hospital stays, which reduce the psychosocial impact of hospitalization.

After being used in adult tumors, the first reported use of LITT for treatment of a pediatric brain tumor was in 2011 for a large thalamic primitive neuroectodermal tumor extending to the midbrain [97]. At three months, the tumor was smaller than pre-ablation levels, and there was no definitive recurrence at 6 months [97]. Since that time, multiple case series and technical reports have been published describing treatment of different locations and types of tumors. Generally, LITT has been used as a primary treatment for deep-seated or eloquent tissue-adjacent tumors for which surgical treatment would be associated with significant operative morbidity. Treatment of thalamic ependymoma, pilocytic astrocytoma, AT/RT, hypothalamic hamartoma, and language and motor-cortex-adjacent gangliogliomas have been described [98,99,100,101]. Ongoing research into the efficacy of primary treatment for many of these low-grade tumors continues.

LITT has also been used for the treatment of radiation necrosis and palliative retreatment of recurrent tumors such as medulloblastoma and pilocytic astrocytoma [99,100]. A multi-institutional retrospective review by Arocho-Quinones et al. included 86 patients with primary and recurrent high- and low-grade tumors. They reported decreased tumor volume in 80%, no need for further surgery in 90%, and stable to improved presenting symptoms in 95% of cases [96].

LITT has a reported complication rate of up to 13% in the pediatric population [102]. Technical complications such as catheter malposition are the most frequently reported at 18% of all complications, followed by neurologic complications at 16% [102]. Cognitive disturbances, seizures, metabolic disturbances, worsening edema, hydrocephalus, shunt malfunction, and hemorrhage have all been described as well [102]. Infection and death are rare but have also occurred. Larger tumor volumes have been associated with a 14% increase in occurrence of complications [96]. Minimally invasive craniotomy and the use of devices such as small tubes or ultrasonic aspirators to resect ablated tissue could help mitigate some effects from post-treatment edema [103]. LITT may also enhance chemotherapy penetration due to disruption of the blood–brain barrier and has been explored with doxorubicin following LITT in newly diagnosed low-grade gliomas [104].

## 4. Conclusions

Surgical advances have made intracranial tumors more accessible, and minimally invasive techniques have improved patient outcomes and quality of life. Research and molecular investigations are providing us with insight into the heterogeneity of pediatric tumors and have informed novel biology-driven therapies, which are a paradigm shift from the conventional chemotherapy regimens which were used in the last century. Multimodal therapy involving laser ablation or convection-enhanced delivery may help overcome some of the challenges in delivering chemotherapy and immunotherapy to the central nervous system. In the future, it is likely that functional imaging and liquid biopsies will play a more prominent role in the molecular diagnosis and monitoring of pediatric CNS tumors and may be all that is needed to diagnose and treat brain tumors.

## Figures and Tables

**Table 1 children-10-00062-t001:** Pediatric Cancer Therapy Consortia.

Consortium	Website
U.S. National Library of Medicine Clinical Trials	https://clinicaltrials.gov/ (accessed on 1 November 2022)
Pacific Pediatric Neuro-Oncology Consortium	https://pnoc.us/ (accessed on 1 November 2022)
Pediatric Brain Tumor Consortium	https://www.pbtc.org/ (accessed on 1 November 2022)
Children’s Oncology Group	https://childrensoncologygroup.org/ (accessed on 1 November 2022)
The DIPG/DMG Resource Network	https://dipg.org/ (accessed on 1 November 2022)
Collaborative Ependymoma Research Network	https://www.cern-foundation.org/ (accessed on 1 November 2022)
Children’s Tumor Foundation	https://www.ctf.org/ (accessed on 1 November 2022)
Children’s Cancer and Leukemia Group	https://www.cclg.org.uk/ (accessed on 1 November 2022)
International Society of Paediatric Oncology	https://siop-online.org/ (accessed on 1 November 2022)

## Data Availability

Not applicable.

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
