# Peer review of "Advances in the Treatment of Pediatric Brain Tumors"

_children, 2022, doi:10.3390/children10010062_

Round 1
Reviewer 1 Report
This paper expose a good overview of pediatric brain tumors and their peculiarity, exspecially their molecular profile. Authors takes more careful consideration molecular targets like therapeutic options but less the various surgical options.
We would suggest a little information about surgical options like instruments of diagnosis and treatment in pediatric brain tumors
Author Response
Point 1: This paper expose a good overview of pediatric brain tumors and their peculiarity, exspecially their molecular profile. Authors takes more careful consideration molecular targets like therapeutic options but less the various surgical options.
We would suggest a little information about surgical options like instruments of diagnosis and treatment in pediatric brain tumors
Response 1: We have added to the section on surgical options to describe the use of our endoscopic surgical tools. We have also changed the title and introduction to highlight the emphasis on molecular profiles and targets.
Reviewer 2 Report
Thank you very much for the opportunity to revise this manuscript.
"Advances in the Diagnosis and Treatment of Pediatric Brain Tumors" focuses as a review article mainly on emerging treatment modalities and treatment advances in pediatric brain tumor patients, taking into account molecular genetic information. In contrast to the "Innovations in Diagnostics" section, the "Advances in Treatment" section includes very detailed descriptions of recent clinical trial results, as well as insights into ongoing prospective studies. The authors should consider focusing exclusively on advances and future perspectives in treatment in this manuscript. The title suggests a more detailed section on "Innovations in Diagnostics."
- 2.1: The introductory sentences are not finally correct. Next to new technologies, the MRI standard sequences are still of upmost importance. Presence of contrast enhancement is no sufficent criterion to descriminate entities or even WHO grades. The T2-FLAIR (abbreviation not introduced) signal is no biomarker (but the T2-signal may be). Radiological tumor assessment has to consider patients age, tumor localization and signal intensities on standard sequences. Please precise the first sentences of the paragraph and add more literature. Furthermore, like it is established for adult brain tumor patients, there are recently published response assessment guidlines for pedriatic tumor entities, which are very important and have to be named there.
- 2.2. Not all cranial GCTs do secrete AFP and beta HCG. Please correct.
- line 472-474: is this a co-author correction?
- there is an extensive paragraph about LITT, but advances in radiotherapy (e.g., proton beam technique) are not addressed
- Reference 8: please correct the style
- Please check for typos in the manuscript (double spaces, dots...)
Author Response
Point 1: "Advances in the Diagnosis and Treatment of Pediatric Brain Tumors" focuses as a review article mainly on emerging treatment modalities and treatment advances in pediatric brain tumor patients, taking into account molecular genetic information. In contrast to the "Innovations in Diagnostics" section, the "Advances in Treatment" section includes very detailed descriptions of recent clinical trial results, as well as insights into ongoing prospective studies. The authors should consider focusing exclusively on advances and future perspectives in treatment in this manuscript. The title suggests a more detailed section on "Innovations in Diagnostics."
That is a fair assessment. We have changed the title to reflect the review more accurately.
Point 2: The introductory sentences are not finally correct. Next to new technologies, the MRI standard sequences are still of upmost importance. Presence of contrast enhancement is no sufficent criterion to descriminate entities or even WHO grades. The T2-FLAIR (abbreviation not introduced) signal is no biomarker (but the T2-signal may be). Radiological tumor assessment has to consider patients age, tumor localization and signal intensities on standard sequences. Please precise the first sentences of the paragraph and add more literature. Furthermore, like it is established for adult brain tumor patients, there are recently published response assessment guidlines for pedriatic tumor entities, which are very important and have to be named there.
This paragraph has been revised to correctly state the role of MRI and standard sequences. We have also addressed the response assessment guidelines that have been created.
Point 2.2. Not all cranial GCTs do secrete AFP and beta HCG. Please correct.
This has been corrected
Point 2.3 line 472-474: is this a co-author correction?
These lines from the MDPI template were removed.
Point 2.4 there is an extensive paragraph about LITT, but advances in radiotherapy (e.g., proton beam technique) are not addressed
A review on radiotherapy techniques and including advantages/disadvantages was thought to be beyond the scope of this paper.
Point 2.5 Reference 8: please correct the style
The reference has been corrected.
Point 2.6 Please check for typos in the manuscript (double spaces, dots...)
The manuscript has been further revised to correct those typos.
Round 2
Reviewer 2 Report
I agree with the revision. Thank you for the opportunity to review this article.
Author Response
Thank-you for your time and assistance.